# OML-AD: ONLINE MACHINE LEARNING FOR ANOMALY DETECTION IN TIME SERIES DATA

## ABSTRACT

Time series are ubiquitous and occur naturally in a variety of applications – from data recorded by sensors in manufacturing processes, over financial data streams to climate data. Different tasks arise, such as regression, classification or segmentation of the time series. However, to reliably solve these challenges, it is important to filter out abnormal observations that deviate from the usual behavior of the time series. While many anomaly detection methods exist for independent data and stationary time series, these methods are not applicable to non-stationary time series. To allow for non-stationarity in the data, while simultaneously detecting anomalies, we propose OML-AD, a novel approach for anomaly detection (AD) based on online machine learning (OML). We provide an implementation of OML-AD within the Python library *River* and show that it outperforms state-of-the-art baseline methods in terms of accuracy and computational efficiency.

## 1 INTRODUCTION

Today's technology ecosystems often rely on anomaly detection for monitoring and fault detection (Ahmad et al., 2017). There are various approaches to anomaly detection (Aggarwal, 2017), but machine-learning (ML) based methods stand out as the most used in real-world use cases (Laptev et al., 2015). Their ability to efficiently process and learn from large datasets led to widespread adoption. However, the general use of classical ML algorithms trained on large batches of data needs to be revised to work for today's dynamically changing and fast-paced systems. The primary concern is the phenomenon of concept drift, which occurs when the statistical properties of the predicted target variable change over time (Lu et al., 2018). As a result, models trained on historical data batches may become outdated, and performance can deteriorate when forecasting (Lu et al., 2018) because of their inability to adapt to changes in the data (Chatfield, 2000). Anomaly detection techniques that rely on accurate predictions of an underlying model suffer from this phenomenon especially. Different approaches to handling concept drift have been proposed in the past (Gama et al., 2014; Lu et al., 2018). One approach is to retrain the model once a change point is detected. While approaches like this can produce satisfactory results, they are complex and costly. Further, they might not detect smooth changes, as occurring in many real-world settings. Hence, there is a need for a robust and dynamic anomaly detection solution that is cheap, performant and able to work with gradual changes.

In this context, online ML emerges as a potential solution. Unlike their batch-learning counterparts, online learning algorithms incrementally perform optimization steps in response to new concepts' influence in the data (Shalev-Shwartz et al., 2012). This continuous learning paradigm enables these algorithms to adapt to changing distributions in data without retraining, thereby ensuring the model's sustained precision. We aim to leverage the features of online learning for predictive anomaly detection on time series data under concept drift to counter common problems of batch-trained ML models.

We propose to combine the existing ideas of prediction-based anomaly detection with online machine learning to create a more dynamic and robust solution.

To compare the proposed approach to similar prediction-based anomaly detection methods commonly employed (e.g., Meta's *Prophet* Taylor & Letham, 2018), we conduct experiments with syn-

thetic and real time series datasets. The benchmark primarily evaluates the accuracy and overall performance of the models, providing a clear comparison of their effectiveness in real-world applications. Besides, additional benchmarks compare both time and resource consumption.

We summarize our contribution as follows:

- We introduce OML-AD, a novel approach to prediction-based anomaly detection using online learning.

- We demonstrate that the proposed approach surpasses state-of-the-art techniques in terms of accuracy, computational efficiency, and resource utilization when handling time series data with concept drift.

- We provide an implementation of our approach within the online machine learning library *River* (Montiel et al., 2021).

## 2 RELATED WORK

The literature on anomaly detection is vast. Two seminal works guide this exploration. Chandola et al. (2009) offer a comprehensive overview of the topic, defining the different types of anomalies, detection methods, and scoring techniques for detection algorithms. Aggarwal (2017) provides an in-depth analysis of different outlier detection methodologies, setting a theoretical baseline for identifying anomalies. In his work, he explains that any ML model used for anomaly detection makes assumptions about the expected behavior of data and uses these expectations to evaluate if a newly seen data point is anomalous.

This statement from Aggarwal lays the foundation for *prediction-based anomaly detection*. With such an approach, a machine learning model learns the normal behavior of a system and makes predictions on newly seen data, to use the prediction error as a metric to identify abnormal behavior. Malhotra et al. (2015) used this paradigm along with Long Short Term Memory Networks to perform anomaly detection on time series. Munir et al. (2018) propose a similar solution called DeepAnT leveraging Convolutional Neural Networks. Liu et al. (2018) use Generative Adversarial Networks to synthetically generate the expected next image of a video and compare it to the actual subsequent frame captured by the camera to detect anomalous activity. Similarly, Laptev et al. (2015) proposed a modular framework for prediction-based anomaly detection called Extensible Generic Anomaly Detection System. The exchangeable modules perform forecasting, anomaly scoring based on the prediction error, and notification on found anomalies.

Time series play an important part in anomaly detection. Blázquez-García et al. (2021) conducted a review of different approaches to anomaly detection on time series specifically. Schmidl et al. (2022) conducted a similar study presenting a wide range of algorithms, which they compare in real-world and synthesized benchmarks, including datasets from the Numenta Anomaly Benchmark. To perform prediction-based anomaly detection on time series data, the base model has to excel at time series forecasting. Chatfield (2000) describes the fundamentals of time series forecasting. One of the most frequently used methods for predicting time series data is Auto-Regressive Integrated Moving Average (ARIMA) modeling, originally proposed by Box and Jenkins (Box et al., 2015).

Traditional models trained on batches of data are susceptible to concept drift, which deserves particular attention in any scenario dealing with a continuous data stream. Lu et al. (2018) examine the problem in detail, illustrate it by example, and suggest ways of detecting it. Similarly, the survey on concept drift adaptation by Gama et al. (2014) deals with the different types of concept drift and suggests multiple ways to adapt.

One possible solution to the problem of concept drift is online learning. Two essential papers on the topic are the article on ML for streaming data by Gomes et al. (2019), and the survey on online learning by Hoi et al. (2021), which both discuss the necessity of online ML and concrete forms of its implementation. With online learning models, training incrementally, a unique form of Gradient Descent, called Online Gradient Descent, is used for optimization, as described by Anava et al. (2013). Similar learning algorithms are used by Guo et al. (2016). They go even further and propose a solution called *adaptive gradient learning*, which makes the learning process robust to outliers but still able to adapt to new normal patterns in the data.

As an alternative to online learning, the quality of ML models might be monitored with methods based on change point detection. With this approach, a model is re-trained whenever a change point in the model's quality is detected. The most common approach, for online change point detection, is based on the CUSUM statistic (see, e. g., Lai, 1995; Chu et al., 1996; Kirch & Stoehr, 2022; Gösmann et al., 2022, among others). In order to prevent the detection of negligibly small changes, different methods have been proposed to detect only relevant changes, that exceed a previously defined threshold (Dette & Wu, 2019; Heinrichs & Dette, 2021; Bücher et al., 2021). For a recent comparison of different monitoring schemes for ML models, see Heinrichs (2023). We will use ADWIN for the batch-trained baseline models in our experiments, which is a commonly used drift detection method, based on sliding windows of adaptive size (Bifet & Gavalda, 2007).

While the majority of research on machine learning for anomaly detection is focused on batch learning techniques, there currently is little effort exploring in online learning for prediction-based anomaly detection. Ahmad et al. (2017) suggest using *Hierarchical Temporal Memory* to continuously learn the behavior of streaming time series data. The online nature of the algorithm automatically handles changes in the underlying statistics of the data. The system models the prediction errors as a Gaussian distribution, allowing for comparing any new error against this distribution. Moreover, Saurav et al. (2018) use RNNs for prediction-based anomaly detection while the core concept of their approach is similar to that of Ahmad et al. (2017). However, Saurav et al. (2018) focus on making their learner robust to outliers while having it adapt to concept drift, a specific problem comparable to the work by Guo et al. (2016).

## 3 PRELIMINARIES

One of the most widely accepted definitions of what an anomaly or an outlier is comes from Hawkins, who describes them as "[...] an observation which deviates so much from the other observations as to arouse suspicions that it was generated by a different mechanism" (Hawkins, 1980). In other words, anomalies are patterns in data that do not conform to the normal behavior of that data, but instead differ from it (Chandola et al., 2009; Schmidl et al., 2022). Anomaly detection is the task of finding such anomalous instances, which can occur as three distinct types: point anomalies, depicted by Figure 1(a), contextual anomalies, and collective anomalies, sometimes also called subsequence anomalies, see Figure 1(b) (Chandola et al., 2009). Whereas point and contextual anomalies occur when the behavior of a single point varies globally (point anomalies) or locally (contextual anomalies), collective anomalies refer to the behavior of multiple points. A particular approach to anomaly detection emerges from a statement by Aggarwal, who wrote that "[...] all outlier detection algorithms create a model of the normal patterns in the data, and then compute an outlier score of a given data point on the basis of the deviations from these patterns" (Aggarwal, 2017). This definition of outliers as values that deviate from expected behavior leads to the idea of prediction-based anomaly detection (Blázquez-García et al., 2021). A well-chosen ML model can learn the normal behavior of a system (Aggarwal, 2017). This model can then predict future behavior, which it considers normal. Comparing the prediction to the actual data point, known as the ground truth, the model can then calculate the anomaly score based on the difference between these two, called the error. Instances that deviate significantly are considered outliers (Blázquez-García et al., 2021). The precision of the underlying model directly correlates with the accuracy of such a detection algorithm (Laptev et al., 2015). Since different models make distinct assumptions about the data, choosing a suitable model is particularly important. If a model cannot represent the normal behavior, this leads to insufficient performance (Aggarwal, 2017).

A specific application area for anomaly detection is the analysis of time series that occur in many places in the industry, e.g., as telemetry data of a monitoring system (Ahmad et al., 2017). To use prediction-based anomaly detection on time series data, the underlying "normal-behavior model" has to be a forecasting model that can predict values of a time series based on historical data by using statistical models to identify patterns and trends. One of the most frequently used models for time series forecasting is the ARIMA model or variations of it (Zhang, 2003). Noted for its flexibility and decent performance, ARIMA is extensively used in diverse real-world scenarios, predicting future values as linear functions of past observations (Zhang & Qi, 2005). The ARIMA model combines an autoregressive (AR) process and a moving average (MA) process. In addition, the original data is "integrated", i. e., replaced by the difference of subsequent observations. The general form of an

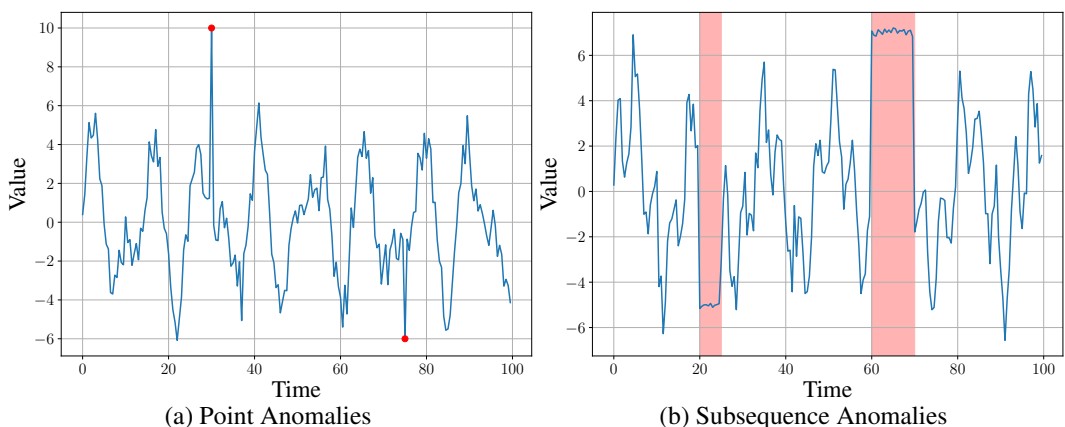

Figure 1: Anomaly Types in Time Series Data

ARIMA$(p, d, q)$ model is given by

$$\Delta^d X_t = \phi_1 \Delta^d X_{t-1} + \phi_2 \Delta^d X_{t-2} + \ldots + \phi_p \Delta^d X_{t-p} + \varepsilon_t + \theta_1 \varepsilon_{t-1} + \theta_2 \varepsilon_{t-2} + \ldots + \theta_q \varepsilon_{t-q},$$

where $\Delta$ denotes the difference operator $\Delta X_t = X_t - X_{t-1}$ and $\Delta^d$ its $d$-fold application. It is a simple regression model that includes the AR and MA components to predict future points. The model might learn the respective coefficients $\phi$ and $\theta$, using maximum likelihood estimation or an optimization algorithm like *Gradient Descent* (Zhang, 2003).

When training such a model to learn the given data's normal behavior, one problem that can occur is concept drift, a phenomenon where the statistical properties of a target variable, which an ML model aims to predict, undergo unexpected changes over time. More precisely, this means there is a change of joint probability of input $X$ and output $y$ at time $t$, denoted by $P_t(X, y)$ (Lu et al., 2018). There is a distinction between *virtual* and *real* concept drift. "The real concept drift refers to changes in the conditional distribution of the output (i.e., the target variable) given the input (input features) while the distribution of the input may stay unchanged" (Gama et al., 2014). Virtual drift, or data drift, on the other hand, refers to a change of $P_t(X)$ only (Lu et al., 2018). Real concept drift can occur in various forms. The two most common forms are sudden and incremental drift (Gama et al., 2014; Lu et al., 2018), which still "[...] correspond to more sustained, long-term changes compared to volatile outliers" (Laptev et al., 2015). Distributions can evolve like this, especially in dynamic data-producing environments that change over time for various reasons, such as hidden changes to the underlying configuration (Gama et al., 2014). This is a problem for model accuracy because the knowledge the model learned from previous data no longer applies to new data, resulting in suboptimal predictions (Lu et al., 2018; Vela et al., 2022). Since these effects on performance are not tolerable for most use cases, scientists developed ways to adapt to this behavior. A straightforward way to do this is to retrain the model on new data regularly (Vela et al., 2022). However, this approach raises the question of when to retrain a model. While doing so on a fixed schedule can work for some use cases, another approach is to retrain a model dynamically using change point detection algorithms like ADWIN (Bifet & Gavalda, 2007). In addition to the conventional method of retraining, techniques such as online learning enable ML models to learn from data one example at a time and adapt to changes in underlying distribution (Lu et al., 2018; Gama et al., 2014).

Online ML models update themselves based on the new distribution of the data (Lu et al., 2018). A continual learning process like this is called online learning or incremental learning. The models update by processing individual instances from a data stream sequentially, one element at a time, performing a forward pass, calculating the loss, and executing a single step of Gradient Descent to update its learnable parameters $\theta$:

$$\theta_i = \theta_{i-1} - \alpha \nabla_{\theta_i} L(\theta_{i-1}).$$

This variation is called Online Gradient Descent (Hoi et al., 2021). It is relatively cheap compared to training on the whole batch, but the update direction will be less precise, which leads to slower or no convergence. However, this circumstance can be good since the model may not get caught

in a local minimum as quickly or overfit the training data (Ketkar, 2017). Online models directly contrast traditional batch-trained ML models, which learn from large datasets that must be available at the beginning of training. On the other hand, online learners can operate without having all the data available right away (Gama et al., 2014). However, single-instance processing has the downside of suboptimal scaling to big data since optimization algorithms cannot use the advantages of vectorization (Montiel et al., 2021). Most online learners assume that the most recent data holds the most significant relevance for current predictions and that a data instance's importance diminishes with age. Therefore, *single example models* store only one example at a time in memory and learn from that example in an error-driven way. They cannot use old examples later in the learning process (Gama et al., 2014). While online learning algorithms usually do not have an explicit forgetting mechanism, like *abrupt forgetting* or *gradual forgetting*, they can still forget old information because the model's parameters update in a way that overwrites or dilutes the knowledge it previously acquired.

## 4 METHODOLOGY

As stated in the introduction, one of our contributions is to develop a solution for prediction-based anomaly detection on time series data under concept drift. While traditionally batch ML has often been used for this kind of application (Malhotra et al., 2015; Laptev et al., 2015; Munir et al., 2018; Liu et al., 2018), some implementations leverage online ML for training models and making predictions as well (Guo et al., 2016; Ahmad et al., 2017; Saurav et al., 2018). Even though these studies lay the groundwork for the new ideas explored in this section, a gap exists in online methods for anomaly detection in time series, especially in applying ARIMA models for forecasting.

The open-source Python library *River* holds a suite of existing tools and models for online learning. Examples include regression models, classification models, clustering algorithms, and forecasting models such as ARIMA's online variant mentioned above. Besides different ML models, the library also offers utilities such as pipelines, tools for hyperparameter tuning, evaluation, and feature engineering, to name a few, specifically designed for online learning (Montiel et al., 2021). Therefore, we present the proposed solution as an additional module for River, called *PredictiveAnomalyDetection* [1], actively enhancing its already available range of features.

We designed the module as a flexible framework to make prediction-based anomaly detection universally applicable across various applications. Choosing the appropriate model to learn the normal behavior of the data is crucial, as an unsuitable choice results in insufficient predictions and, therefore, low detection accuracy. What is the best fitting model depends on the underlying data and associated assumptions (Aggarwal, 2017). Therefore, the underlying model for learning normal behavior is not set in the module but can be defined when initializing a new detector instance. This design adds versatility, allowing users to choose from various online learning models available within River. For the problem stated in this work, the online ARIMA variant (Anava et al., 2013) plugs into this framework to detect point and contextual anomalies in time series data.

The chosen design conceptually separates the modeling of expected behavior from the scoring process, similar to the approach used by Laptev et al. (2015). The base estimator predicts the expected behavior of the data and compares it to the actual value to calculate the error. The detection algorithm uses this error value, independently of the base estimator it came from, to calculate the anomaly score.

The scoring mechanism involves comparing the prediction with the ground truth, where deviation signifies error and, consequently, the score. More specifically, if $X_t$ denotes the true value of a time series at time $t$ and $\hat{X}_t$ is an estimator of it, based on past values $(X_i)_{i<t}$, then the error is defined as $\hat{\varepsilon}_t = |\hat{X}_t - X_t|$. For some threshold $\tau > 0$, the score $s_t$ of $\hat{\varepsilon}_t$ is defined as

$$s_t = \min\left\{\frac{\hat{\varepsilon}_t}{\tau}, 1\right\}, \tag{1}$$

which takes values between 0 and 1, and a score of 1 strongly indicates an anomaly. The choice of the threshold $\tau$ plays a crucial role in the definition of an outlier. The simplest choice is to use $\tau_0 = \mu_t + c\sigma_t$, where $\mu_t$ and $\sigma_t^2$ denote the (possibly time-dependent) mean and variance of the errors and $c$ a constant, specifying the sensitivity towards anomalies.

---

[1] The code is in the official repository for river: non-anonymized link in final version of this paper.

Another approach is based on the common assumption that the residuals $\hat{X}_t - X_t$ are independent and (approximately) normally distributed with variance $\sigma_t^2$, i.e., $\hat{X}_t - X_t \sim \mathcal{N}(0, \sigma_t^2)$. In the simplest case, $\sigma = \sigma_t$ is constant over time, yet analogous arguments are valid in the contrary case. Let $\hat{\sigma}$ be a consistent estimator of $\sigma$, then $\hat{\varepsilon}_t / \hat{\sigma}$ has (approximately) the distribution $|\mathcal{N}(0, 1)|$. Let $q_{1-\alpha}$ denote the $(1-\alpha)$-quantile of the distribution $|\mathcal{N}(0, 1)|$, for $\alpha \in (0, 1)$, then we can define $\tau_1 = q_{1-\alpha} \hat{\sigma}$ as threshold for the score $s_t$. With this choice, we have a probability of falsly detecting an anomaly of $\alpha$, for each time point $t \in \mathbb{N}$.

If the latter probability is too high for our application, we can use extreme value theory to find a more conservative choice of $\tau$. Note that the (appropriately scaled) maximum over normally distributed random variables converges weakly to a Gumbel distribution. More specifically, let $a_n = \sqrt{2\log(2n)}$, $b_n = a_n^2 - \frac{1}{2}\log(4\pi \log(2n))$ and $Z_1, \ldots, Z_n$ be independent random variables with distribution $|\mathcal{N}(0, 1)|$. It is well known that

$$\lim_{n \to \infty} P(a_n \max_{i=1}^{n} Z_i - b_n \leq x) = \exp(-\exp(-x))$$

(Leadbetter et al., 2012). Alternatively to selecting $\tau$ based on quantiles of $|\mathcal{N}(0, 1)|$, we might as well set $\tau_2 = \{(q'_{1-\alpha} + b_n)\hat{\sigma}\}a_n^{-1}$, where $q'_{1-\alpha} = -\log(-\log(1-\alpha))$ denotes the $(1-\alpha)$ of the standard Gumbel distribution, for $\alpha \in (0, 1)$. With this choice, we can (asymptotically) bound the probability of a false positive in $n$ sequential residuals by $\alpha$. Clearly, with this conservative choice of $\tau$, it is more likely that some anomaly gets a score less than 1 compared to the choice $\tau_1$.

## 5   EMPIRICAL FINDINGS

*Datasets.* Using high-quality datasets is crucial to accurately evaluate the performance of the proposed method. However, many publicly available time series datasets suffer from unrealistic anomaly density or incorrect labeling of ground truth values (Wu & Keogh, 2021). As online learning is particularly relevant when the distribution of the data generating process changes over time, the considered datasets should contain some form of drift. We considered three different datasets for our experiments.

First, we generated synthetic data with a varying mean and a specified number of anomalies. For a time horizon of $n = 1\,000$ steps, we generated "normal" time series $X_t = \sin(30\pi \frac{t}{n}) - (3\frac{t}{n} - 1)^2 + \varepsilon_t$, where $\varepsilon_t \sim \mathcal{N}(0, \frac{1}{4})$ denote independent random variables, for $t = 1, \ldots, n$. We added randomly "anomalies" to 5% of the generated observations with a random height, sampled uniformly from $[-2, -1] \cup [1, 2]$. Figure 2 shows an exemplary trajectory of the generated time series. With this approach, we generate point anomalies, with values that differ from the global behavior of the time series, and contextual anomalies, that only deviate from the local behavior. Clearly, the latter are harder to detect.

Second, we used weather data from various Australian cities spanning approximately 150 years, which can be considered as non-stationary (Bücher et al., 2020). To create realistic anomalies within this dataset, we synthesized them by mutating some temperature recordings from degrees Celsius to degrees Fahrenheit. Figure 3 shows the prepared data. Additionally, to complement our evaluation and incorporate real-world data, we utilized the CPU load data from a cloud instance provided by the Numenta Anomaly Benchmark. Despite its smaller scope, this dataset offered a valuable perspective by providing a realistic environment for the benchmarks.

*Metrics.* We conduct three benchmarks to assess the accuracy of the proposed approach compared to baseline models[2]. The first experiment evaluates time series forecasting as well as anomaly detection performance. Accurate forecasting leads to better anomaly detection, as more significant deviations between predicted and actual values indicate anomalies (Laptev et al., 2015). This benchmark measures the Mean Absolute Error (MAE) and Mean Squared Error (MSE) to assess forecasting accuracy. Further, we use the F1 score and the ROC AUC to evaluate anomaly detection performance. The predicted anomaly scores are converted to binary labels to calculate these metrics using thresholds optimized for each model's F1 score, ensuring fair comparisons. The second benchmark tracks CPU and RAM usage during a fixed period, while the third experiment measures the time each model requires for training, prediction, and anomaly scoring. Each benchmark is repeated

---

[2]The code for benchmarking can be found here: non-anonymized link in final version of this paper

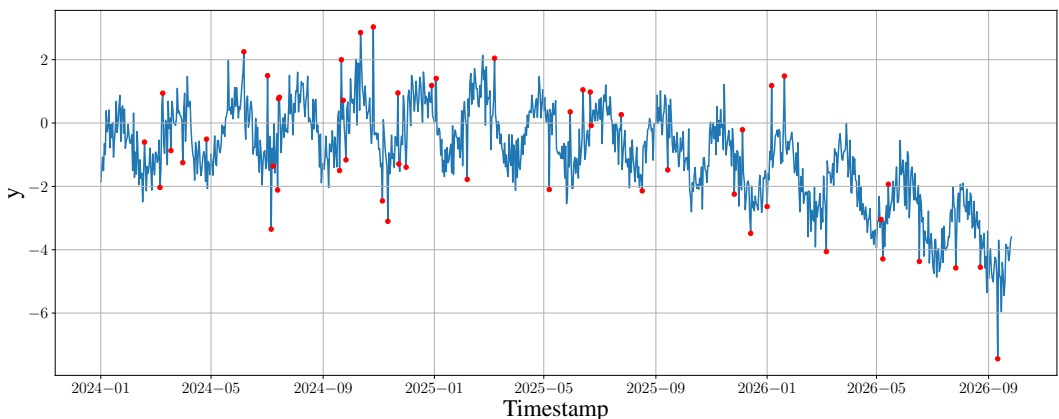

Figure 2: Synthesized Dataset

100 times, with results averaged for accuracy. To ensure comparability, all tests are conducted on the same dedicated virtual machine within a Docker container, minimizing external influences on performance.

*Baseline Models.* We compare the proposed OML-AD module with two baseline models. The baselines are the SARIMA model from the *statsmodels* library and Meta's *Prophet* model (Taylor & Letham, 2018). We adapted both models as time series forecasting tools for prediction-based anomaly detection. Hyperparameters for all models were manually tuned for optimal performance, though we excluded this process from time and resource consumption benchmarks.

The first baseline, SARIMA, is a traditional batch model for time series forecasting. Anomaly scores are calculated based on the model's error distribution. Similar to the OML-AD module, anomalies are identified by significant deviations between the predicted values and the actual observations. The SARIMA model was configured with optimal parameters for this use-case, identified using the *pmdarima* library: $(p, d, q) = (1, 0, 1)$ and $(P, D, Q, s) = (1, 1, 1, 52)$ or $(s = 24,$ for the NAB CPU utilization data). The model was optimized using the default maximum likelihood estimation via the Kalman filter, as implemented in the *statsmodels* library.

*Prophet*, the second batch-trained baseline model, is recognized for its speed and simplicity (Taylor & Letham, 2018). Like SARIMA, it trains on a fixed amount of data, with anomaly detection relying on the error distribution to identify deviations. The Prophet model was used with default settings, except for explicitly enabling yearly and weekly seasonality.

To comprehensively evaluate the models, we implement three retraining strategies. The first is fixed schedule retraining, where models periodically retrain using a fixed amount of the most recent data (in our case every 800 entries), simulating a sliding window approach. The second strategy involves dynamic retraining, utilizing change point detection through ADWIN (Bifet & Gavalda, 2007) to identify shifts in data distribution, prompting the model to retrain on the latest data. We employed ADWIN with the specific parameters $delta = 0.001$, $max\_buckets = 10$, $grace\_period = 10$, $min\_window\_length = 10$, $clock = 20$. Lastly, we simulate the traditional batch method, where models are trained once on the initial training set, consisting of the first 800 entries, and remain static throughout the experiment.

Our proposed OML-AD module, introduced before, differs by using an online learning approach, updating its parameters continuously with incoming data. For the conducted benchmarks, it employs an online SARIMA variant as its underlying forecasting model. We selected the threshold $\tau$ from equation 1 as $\mu_t + 3\sigma_t$, where $\mu_t$ and $\sigma_t$ were updated based on recent observations Chandola et al. (2009). The model utilizes *rivers* SNARIMAX model as its base, configured with the following set parameters: $(p, d, q) = (2, 1, 2)$ and $(P, D, Q, s) = (2, 0, 2, 52)$ or $(s = 24,$ for the NAB CPU utilization data). The model's learned parameters are optimized using Stochastic Gradient Descent with a learning rate of $0.001$, after preprocessing with a *StandardScaler*.

*Results.* Detailed results from the experiments with synthetic data are displayed in Tables 1 and 2, while the results of the other datasets can be found in the appendix. The first benchmark assessed the forecasting performance of the different models using the MAE and MSE. Contrary to the expectation that batch models would outperform OML-AD due to their ability to leverage the entire dataset for parameter estimation, the latter demonstrated superior performance with lower MAE and MSE values than the baseline with no retraining. This difference in overall forecasting performance is likely because the online model's continuous adaptation allowed it to handle the abrupt concept drift better. In contrast, batch models struggled to adapt to changes in the data.

Figure 4 and Figure 5 illustrate this behavior. SARIMA fails to adapt to concept drift, resulting in noticeable shifts in forecast errors (see Figure 7). As a result, OML-AD outperforms the batch models in terms of F1 score and AUC-ROC. In light of these findings, it is evident that while batch learning methods like SARIMA perform well in stable environments, they falter in the presence of concept drift compared to online learning approaches. This discrepancy highlights the inherent limitations of batch learning in dynamically changing environments. We conclude that the proposed online learning approach offers superior accuracy in prediction-based anomaly detection on time series data under concept drift, demonstrating its effectiveness and robustness in evolving conditions. Specifically for the synthesized data with contextual anomalies, that are generally hard to detect, OML-AD has a substantially higher F1 score and AUC ROC than the considered alternatives.

While retraining batch models can theoretically address concept drift, it remains unclear whether an online learning approach is more resource-efficient and time-effective. Our benchmarks, which included both scheduled and dynamic retraining for batch models, revealed that retraining significantly improves their performance, sometimes even matching that of the online model. Notably, dynamic retraining proved more effective than fixed scheduled retraining, with its success depending on the underlying drift detection algorithm. In contrast, the effectiveness of scheduled retraining is contingent on the chosen schedule or window size.

Despite these improvements, OML-AD still demonstrates superior computing power and memory usage efficiency. The CPU usage benchmark shows that OML-AD requires the least computing power on average, while the memory usage benchmark indicates that OML-AD allocates less RAM than SARIMA and Prophet. However, all models exhibit relatively even memory consumption overall. OML-AD's efficiency comes from its online gradient descent algorithm, which processes data one example at a time. This approach minimizes memory usage by eliminating the need to load the entire dataset simultaneously, and reduces the computational cost of each individual gradient descent step. Timing benchmarks also reveal that OML-AD consistently outperforms batch models in terms of speed due to the low computational cost of its operations. However, it is essential to note that an online model like OML-AD must remain active to receive incoming data, which, although often idle, still occupies some resources. Additionally, when batch models employ scheduled or dynamic retraining, they become even slower, further widening the performance gap. This trade-off highlights the complexity of balancing resource efficiency and model performance in dynamically changing environments.

Table 1: Forecasting and detection on synthetic data

| Algorithm | | MAE | MSE | F1 | AUC ROC |
|---|---|---|---|---|---|
| OML-AD | | **0.5221** | **0.5008** | **0.7619** | **0.9768** |
| SARIMA | No Retraining | 1.9629 | 6.5499 | 0.2803 | 0.2803 |
| | Scheduled Retraining | 1.0367 | 1.7046 | 0.4622 | 0.7107 |
| | Dynamic Retraining | 0.9452 | 1.4367 | 0.5320 | 0.7467 |
| Prophet | No Retraining | 2.9476 | 12.8837 | 0.0949 | 0.4898 |
| | Scheduled Retraining | 1.3307 | 2.9898 | 0.1362 | 0.6401 |
| | Dynamic Retraining | 1.7373 | 4.7683 | 0.2214 | 0.6713 |

Table 2: Time and resource consumption on synthetic data

| Algorithm | | Mean Time [ms] | Std [ms] | CPU [%] | RAM [%] |
|---|---|---|---|---|---|
| OML-AD | | **38.99** | **7.02** | **3.09** | **27.84** |
| SARIMA | No Retraining | 29368.83 | 18720.22 | 7.50 | 43.84 |
| | Scheduled Retraining | 155327.29 | 31067.93 | 7.17 | 39.23 |
| | Dynamic Retraining | 83937.51 | 18196.82 | 11.01 | 40.36 |
| Prophet | No Retraining | 259.63 | 66.00 | 4.72 | 36.20 |
| | Scheduled Retraining | 1155.41 | 158.97 | 4.97 | 31.41 |
| | Dynamic Retraining | 833.46 | 192.60 | 4.99 | 36.80 |

## 6 LIMITATIONS

Several limitations are inherent in the methodology used in this study. We conducted the measurements and benchmarks using synthetic data, weather data with synthesized anomalies and real CPU load data from the Numenta Anomaly Benchmark. The data with synthezied anomalies, while controlled, may not fully capture the complexity of real-world scenarios. The CPU load data, on the other hand, contains few anomalies, which complicates performance evaluation and can affect the reliability of the metrics. While these datasets provide diverse scenarios, limitations arise due to the focus on specific use cases. Furthermore, this data includes only specific types of concept drift and particular anomaly types, namely point and contextual anomalies. Future research could address this limitation by exploring consecutive anomalies and using a predictive model-based approach along with longer forecasting horizons (Blázquez-García et al., 2021). Besides, the accuracy of prediction-based anomaly detection depends on the suitability of the underlying model to the use case and the data, emphasizing the need for precise tailoring. In this paper, we focused only on three specific use cases, which presents a challenge in terms of generalizability.

A significant challenge identified in this study is distinguishing between concept drift and outliers, which is particularly critical in anomaly detection. Abrupt changes may resemble anomalies, while gradual drift might be less identifiable, blurring the line between the two. The *Adaptive Gradient Learning* method presented by Guo et al. (2016) is an approach to counter this problem. Though innovative, it is not infallible and requires extensive testing across different scenarios. Guo et al. (2016) found that this approach is more effective when predicting multiple steps, but it relies on multiple ground truth values, causing a delay in detection (Guo et al., 2016; Saurav et al., 2018). The distinction between concept drift and outliers remains a complex challenge in anomaly detection, necessitating careful consideration of the model's response to various types of drift and the potential integration of specific tests and strategies to enhance adaptability and accuracy.

Another aspect not fully addressed in this paper is hyperparameter tuning. The benchmarks focused solely on training, inference time, and resource consumption, omitting hyperparameter optimization for fair comparison. However, hyperparameter tuning is essential to the machine learning lifecycle in real-world applications, often managed through MLOps practices (Sculley et al., 2015; Mäkinen et al., 2021; Kreuzberger et al., 2023). While online learning offers a solution to concept drift by reducing the need for frequent retraining, it introduces specific challenges that MLOps must address. Parameter-laden algorithms, especially in online environments, require delicate tuning, as they can be susceptible to parameter settings like internal thresholds or learning rates (Laxhammar & Falkman, 2013). Traditional tuning methods, such as grid or random search, are not readily applicable in online settings (Gomes et al., 2019). Moreover, adapting the fundamental structure of a model, such as altering ARIMA parameters ($p$, $d$, $q$), may be necessary depending on system behavior. MLOps is crucial in addressing these challenges, encompassing hyperparameter tuning, continuous model performance monitoring, rollback capabilities, and efficient deployment strategies. However, most MLOps frameworks focus on classical batch learning setups and often overlook the unique challenges online learning poses.

## 7 CONCLUSION

The findings from this research have significant practical implications, particularly for industries reliant on real-time data analysis. The demonstrated superiority of OML-AD in specific settings highlights its efficiency as a solution for online anomaly detection in the presence of concept drift. Implementing an online ML model, like OML-AD, for prediction-based anomaly detection offers distinct advantages over traditional batch-learning approaches. OML-AD's online learning capability is particularly suited for real-time applications, enabling continuous processing of data streams without the need for periodic retraining. This adaptability is crucial in industries where non-stationarity of data is common, as OML-AD can handle unpredictable changes in distributions and trends more effectively than batch-learning methods. In such dynamic environments, OML-AD's ability to continuously adapt to new data patterns without requiring retraining makes it an invaluable tool, especially where timely and accurate anomaly detection is critical and retraining larger batch-trained models regularly is impractical (Gama et al., 2014).

Our formulation of the anomaly score in equation 1 allows for the definition of theoretically sound anomalies, our empirical results showed that OML-AD is superior or similar to the considered alternatives in terms of MAE, MSE, F1-score and AUC ROC. Further, it used significantly less memory and time compared to the baseline models. Thus, for settings similar to the evaluated datasets, the proposed method, based on the SARIMA model, is recommended. In more complex situations, the SARIMA model might be replaced by a different model that fits the "normal" data well.

While this work focused on point and contextual outliers, it remains open to study how the proposed method can be adjusted to collective anomalies and how well it compares to other methods for this specific types of anomalies.

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

Table 3: Forecasting and detection performance on weather data with synthesized anomalies

| City | Algorithm | | MAE | MSE | F1 | AUC ROC |
|------|-----------|---|-----|-----|-----|---------|
| Sydney | OML-AD | | 2.7504 | **8.0843** | **0.9503** | 0.9879 |
| | SARIMA | No Retraining | 6.3630 | 69.0261 | 0.1320 | 0.9765 |
| | | Scheduled Retraining | 2.5258 | 21.9888 | 0.6170 | 0.9861 |
| | | Dynamic Retraining | **2.4962** | 20.9147 | 0.8862 | **0.9968** |
| | Prophet | No Retraining | 16.3098 | 387.5487 | 0.0398 | 0.8558 |
| | | Scheduled Retraining | 6.5243 | 68.9949 | 0.7420 | 0.9651 |
| | | Dynamic Retraining | 2.5856 | 23.6932 | 0.8025 | 0.9677 |
| Melbourne | OML-AD | | 2.7637 | **7.9064** | **0.9747** | **0.9998** |
| | SARIMA | No Retraining | 6.0970 | 66.6365 | 0.1370 | 0.9584 |
| | | Scheduled Retraining | 2.5819 | 22.5574 | 0.5987 | 0.9957 |
| | | Dynamic Retraining | **2.4129** | 21.1346 | 0.9014 | 0.9989 |
| | Prophet | No Retraining | 17.0762 | 404.8067 | 0.0402 | 0.8425 |
| | | Scheduled Retraining | 6.6228 | 69.2407 | 0.7230 | 0.9975 |
| | | Dynamic Retraining | 2.6378 | 23.7981 | 0.8318 | 0.9987 |
| Robe | OML-AD | | 2.6104 | **7.5719** | **0.9719** | **0.9988** |
| | SARIMA | No Retraining | 6.4173 | 68.7132 | 0.1372 | 0.9490 |
| | | Scheduled Retraining | 2.5203 | 23.6533 | 0.5857 | 0.9942 |
| | | Dynamic Retraining | 2.5432 | 20.1169 | 0.8599 | 0.9939 |
| | Prophet | No Retraining | 18.0550 | 397.8834 | 0.0389 | 0.8043 |
| | | Scheduled Retraining | 6.6134 | 66.2162 | 0.7011 | 0.9454 |
| | | Dynamic Retraining | **2.4831** | 24.0231 | 0.8046 | 0.9621 |

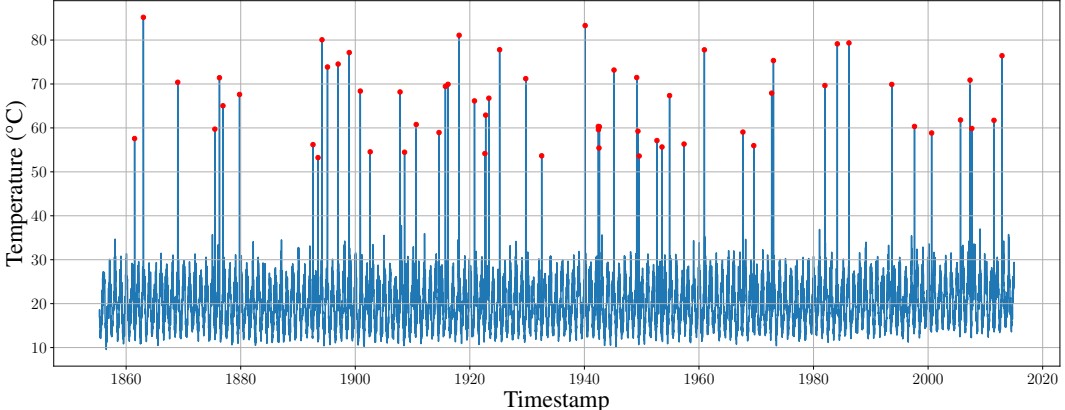

Figure 3: Weekly Temperature Data with Synthesized Anomalies

# A  APPENDIX

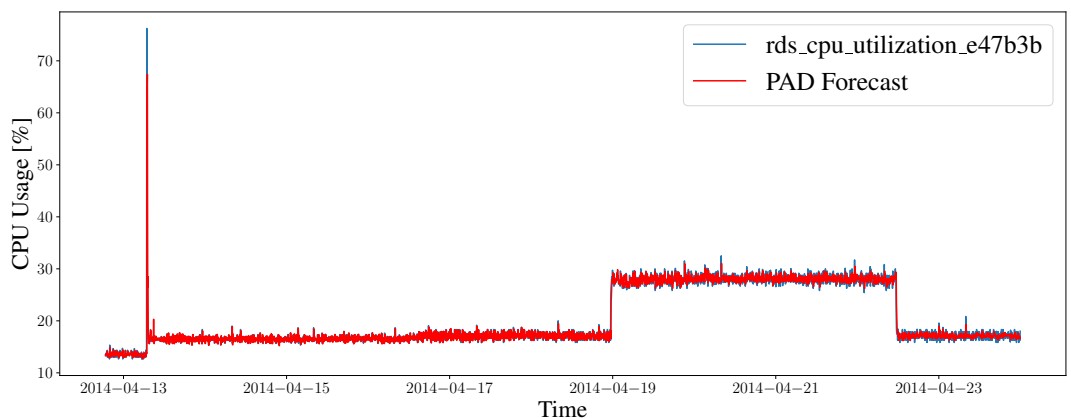

Figure 4: OML-AD Forecast on CPU Utilization Data

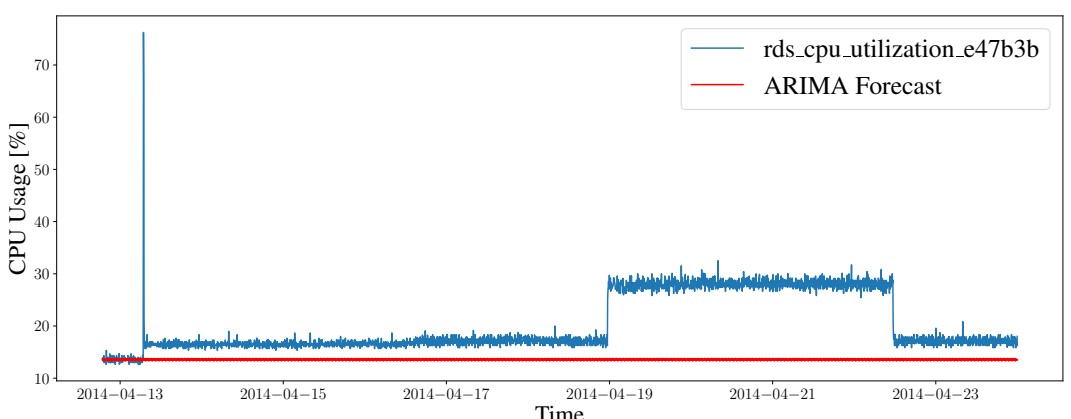

Figure 5: SARIMA Forecast on CPU Utilization Data without Retraining

Table 4: Time and resource consumption on weather data with synthesized anomalies

| City | Algorithm | | Mean Time [ms] | Std [ms] | CPU [%] | RAM [%] |
|------|-----------|--|----------------|----------|---------|---------|
| Sydney | OML-AD | | **628.83** | **364.26** | **3.95** | **22.09** |
| | SARIMA | No Retraining | 58913.33 | 774.78 | 15.05 | 29.52 |
| | | Scheduled Retraining | 164313.09 | 2098.46 | 15.83 | 29.10 |
| | | Dynamic Retraining | 344827.70 | 4911.51 | 15.23 | 30.57 |
| | Prophet | No Retraining | 2078.27 | 459.57 | 4.20 | 33.21 |
| | | Scheduled Retraining | 6482.84 | 2669.68 | 12.91 | 29.42 |
| | | Dynamic Retraining | 12132.69 | 1178.52 | 11.95 | 29.21 |
| Melbourne | OML-AD | | **660.78** | **352.42** | **4.16** | **23.00** |
| | SARIMA | No Retraining | 57674.96 | 741.55 | 14.78 | 30.92 |
| | | Scheduled Retraining | 164430.49 | 2013.63 | 15.79 | 30.01 |
| | | Dynamic Retraining | 340427.43 | 4686.40 | 15.32 | 29.71 |
| | Prophet | No Retraining | 2173.90 | 460.67 | 4.22 | 31.76 |
| | | Scheduled Retraining | 6445.34 | 2715.17 | 13.33 | 30.41 |
| | | Dynamic Retraining | 12274.80 | 1149.53 | 11.78 | 29.84 |
| Robe | OML-AD | | **699.30** | **353.24** | 4.16 | **21.86** |
| | SARIMA | No Retraining | 60707.34 | 781.29 | 15.52 | 32.69 |
| | | Scheduled Retraining | 155240.83 | 1941.19 | 15.93 | 28.45 |
| | | Dynamic Retraining | 342832.87 | 4883.07 | 14.80 | 29.97 |
| | Prophet | No Retraining | 2171.43 | 445.04 | **4.06** | 31.07 |
| | | Scheduled Retraining | 6506.64 | 2823.17 | 13.42 | 29.28 |
| | | Dynamic Retraining | 11824.89 | 1115.07 | 11.36 | 30.04 |

Table 5: Forecasting and detection performance on CPU utility data with real anomalies

| Algorithm | | MAE | MSE | F1 | AUC ROC |
|-----------|--|-----|-----|-----|---------|
| OML-AD | | **0.7525** | **2.4217** | 0.4444 | **0.9992** |
| SARIMA | No Retraining | 6.7164 | 75.5092 | **0.5000** | 0.8438 |
| | Scheduled Retraining | 4.2726 | 39.9807 | **0.5000** | 0.8420 |
| | Dynamic Retraining | 1.2050 | 5.9659 | 0.0615 | 0.9906 |
| Prophet | No Retraining | 8.0303 | 99.8151 | **0.5000** | 0.8438 |
| | Scheduled Retraining | 3.9737 | 29.1455 | **0.5000** | 0.8686 |
| | Dynamic Retraining | 10.0246 | 470.6927 | 0.0190 | 0.7545 |

Table 6: Time and resource consumption on CPU utility data with real anomalies

| Algorithm | | Mean Time [ms] | Std [ms] | CPU [%] | RAM [%] |
|-----------|--|----------------|----------|---------|---------|
| OML-AD | | **154.96** | **7.04** | 2.82 | **31.38** |
| SARIMA | No Retraining | 6074.72 | 1128.20 | 6.13 | 48.11 |
| | Scheduled Retraining | 43000.47 | 6034.02 | 9.71 | 41.56 |
| | Dynamic Retraining | 31035.75 | 3293.89 | 9.99 | 39.81 |
| Prophet | No Retraining | 592.08 | 33.82 | **2.39** | 42.04 |
| | Scheduled Retraining | 2194.62 | 579.22 | 9.04 | 41.18 |
| | Dynamic Retraining | 4442.64 | 260.73 | 7.09 | 41.43 |

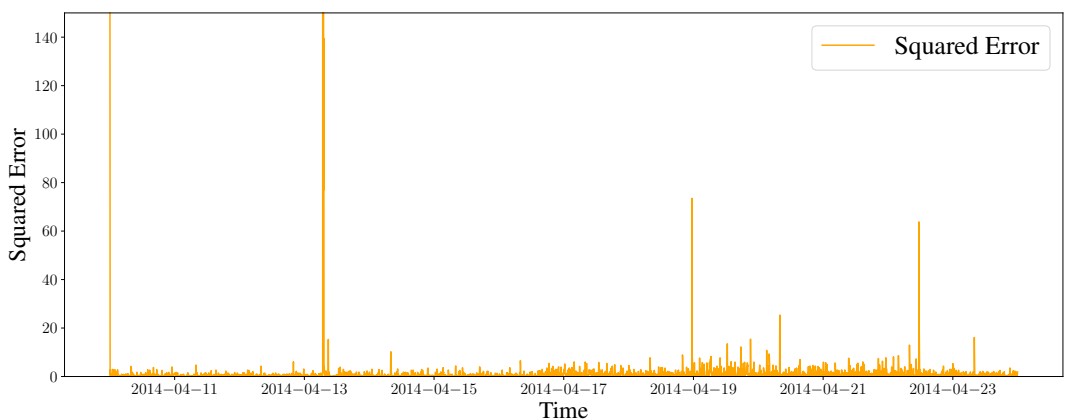

Figure 6: Error of OML-AD Forecast on CPU Utilization Data

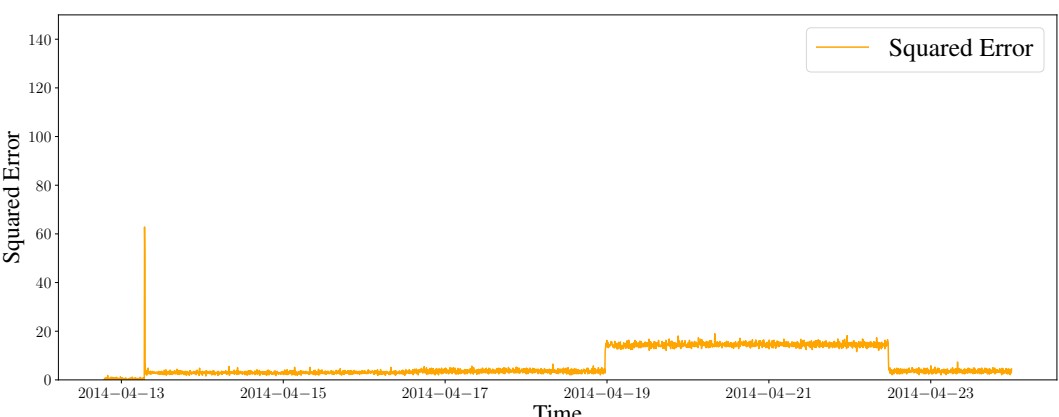

Figure 7: Error of SARIMA Forecast on CPU Utilization Data

