# OpenReview forum: "OML-AD: Online Machine Learning for Anomaly Detection in Time Series Data"
_ICLR.cc/2025/Conference — Submitted to ICLR 2025_

### Official Review · Reviewer_3SaC · 2024-10-26

**Soundness:** 1
**Presentation:** 1
**Contribution:** 2
**Rating:** 3
**Confidence:** 4

**Summary:**

This paper introduces an online learning method for prediction-based anomaly detection. The proposed method can better handle the data drift in an online setting than batch-trained models.

This paper has bad writing, and the contribution is limited.

**Strengths:**

1. The method is integrated into the widely-used library River.
2. The proposed method is simple.

**Weaknesses:**

1. The writing is horrible. The whole paper needs to be rewritten completely. The introduction, related work, and preliminary seem to be redundant, and the key points cannot be reflected clearly.

2. The contribution is limited. It is hard to find the contribution. It is hard to understand how the proposed method is relevant to online learning in the main content.

3. The experiment is weak. Why not use KDD cup datasets? There are 250 time series in the KDD cup. Also, a robust method should be flexible to handle different data drift, including no drift, slow drift, fast drift, and random drift, as the non-stationarity in the real world is diverse in different applications.

**Questions:**

There are too many questions to ask. To address them, the paper has to be rewritten completely.

---

### Official Review · Reviewer_5q9q · 2024-10-31

**Soundness:** 1
**Presentation:** 2
**Contribution:** 1
**Rating:** 3
**Confidence:** 4

**Summary:**

The paper proposes an online anomaly detection method capable of handling non-stationary time series data through predictive approaches. A limited number of experiments were conducted to verify the effectiveness of the method.

**Strengths:**

The proposed method is capable of handling time series data with concept drift.

**Weaknesses:**

The issues highlighted in the paper are not novel, as time series prediction is a well-researched problem. Building upon this foundation, only minor modifications to models are required to perform prediction-based time series anomaly detection, contrary to the paper's claim that "there currently is little effort exploring online learning for prediction-based anomaly detection."

The experimentation is insufficient:
1. The comparative algorithms used are from 2018 and only two such algorithms are considered.
2. There is a lack of crucial experiments, such as ablation studies.

**Questions:**

Additional experiments could be conducted to further validate the results.

---

### Official Review · Reviewer_W1HX · 2024-11-01

**Soundness:** 1
**Presentation:** 2
**Contribution:** 1
**Rating:** 3
**Confidence:** 4

**Summary:**

The paper mainly proposes OML-AD (Online Machine Learning for Anomaly Detection), which is a flexible framework that can use different online forecasting models as base models to learn normal behavior. It then determines whether an observation is an anomaly by comparing the prediction errors with a threshold, which is dynamically changing. This approach addresses the issue of models trained on historical data becoming outdated and their inability to adapt to changes in the data.

**Strengths:**

S1. This paper introduces a lightweight plugin that effectively transforms a basic online forecasting model into an online anomaly detection model. This innovative approach enhances the adaptability of existing methods that are trained on historical data.

S2. The writing in the paper is generally clear and well-structured, effectively communicating the design of the OML-AD framework.

S3. This paper addresses a critical issue in anomaly detection: the inability of models trained on historical data to adapt to changes in future data. By tackling this problem, the research holds practical significance for real-world applications.

**Weaknesses:**

W1. I think the overall approach has too little workload and lacks innovation, as it simply uses the prediction error between the predicted values of an existing online forecasting model and the actual values, with a threshold to determine if it’s an anomaly. Even the method for updating μt and σt, which are key to calculating the threshold, is based on previous studies.

W2. The experiments conducted in the paper primarily utilize a narrow range of datasets, including one synthetic dataset and one real-time dataset with added synthetic anomalies. The authors should consider adding more real-world datasets, particularly those with varying degrees of concept drift and different types of anomalies.

W3. While the paper compares the proposed approach to a few existing methods, the selection of baseline models appears limited. For a more robust evaluation, the authors should include a wider variety of state-of-the-art anomaly detection techniques.

**Questions:**

Q1. I would like to inquire whether the MAE and MSE performance of OML-AD is solely dependent on the models used within it. From my understanding, OML-AD does not incorporate specific measures to impove MAE and MSE performance. Therefore, I believe that to effectively demonstrate the performance of OML-AD, it should be compared with models having similar MAE and MSE performance.

Q2. In the 4 METHODOLOGY section of the article, three methods for calculating threshold are mentioned: simple choice using  μt and σt, based on the common assumption of residuals' normality and using extreme value theory to find a more conservative threshold. However, only the first method was used in the experiments. I think corresponding experiments should be added to showcase their application scenarios.

Q3. To showcase the effectiveness of OML-AD, it would be beneficial to include comparisons with several recent state-of-the-art anomaly detection models, rather than relying solely on SARIMA and Prophet, which were originally designed for forecasting but have been repurposed for anomaly detection.

Q4. While the synthetic dataset in Figure 2 shows some degree of concept drift, I believe that recent state-of-the-art anomaly detection models can handle such scenarios well. Could the authors provide examples of situations that are particularly challenging for other models but can be effectively addressed by OML-AD?

---

### Official Review · Reviewer_vQrN · 2024-11-05

**Soundness:** 2
**Presentation:** 2
**Contribution:** 1
**Rating:** 3
**Confidence:** 4

**Summary:**

The paper presents OML-AD, an online anomaly detection method for non-stationary time-series data. The method combines existing ideas scatter in literature for different tasks and demonstrate the potential of the solution on several datasets.

**Strengths:**

S1. Online anomaly detection is critical for the massive amounts of data, streams, and edge devices.
S2. Good coverage of methods in online learning broadly
S3. Results support the overall claim

**Weaknesses:**

W1. Lack of technical depth
W2. Missing baselines
W3. Missing progress in the area for benchmarks
W4. Missing progress in the area for evaluating anomaly detectors

**Questions:**

W1. Lack of technical depth

The paper presents a solution that is mainly a study of existing solutions combined for this task. Therefore, the technical depth is somewhat low (even though the combination of such ideas in general might be novel)

W2. Missing baselines

Overall, the coverage of online learning is good. Unfortunately, the work is missing specialized solutions for the time-series anomaly detection task. For example [a] solves a very similar problem so it should be used, along with other methods in [a], as baselines.

[a] "SAND: streaming subsequence anomaly detection." Proceedings of the VLDB Endowment 14.10 (2021): 1717-1729.

W3. Missing progress in the area for benchmarks

Similarly, there has been tremendous progress in benchmarking anomaly detectors [b]

[b] "TSB-UAD: an end-to-end benchmark suite for univariate time-series anomaly detection." Proceedings of the VLDB Endowment 15.8 (2022): 1697-1711.

W4. Missing progress in the area for evaluating anomaly detectors

As with previous comments, the work is missing progress towards evaluating methods in this area [c]

[c] "Volume under the surface: a new accuracy evaluation measure for time-series anomaly detection." Proceedings of the VLDB Endowment 15.11 (2022): 2774-2787.

---

### Meta-Review · Area_Chair_fhkx · 2024-12-17

**Metareview:**

The paper presents OML-AD, an online method for anomaly detection that aims to handle non-stationary time series by adapting a forecasting model on the fly. While it tries to address the concept drift problem, the actual improvements shown over existing baselines seem pretty modest.

There is no response from the authors.

At the end of the discussion phas, all reviewers still aligned on a negative stance. No one was convinced that the current evaluation or experimental design demonstrated a solid advantage. Instead, they all remained critical of the limited range of datasets and older baselines. Two reviewers explicitly mentioned the need for more modern and relevant comparison methods, and at least two pointed out the experiments should have included more challenging real-world scenarios beyond just the synthetic or very limited sets tried here.

Before this work could be accepted, it would need stronger, more up-to-date baselines and a more diverse set of datasets to show OML-AD’s effectiveness. Without those improvements, it is difficult to have confidence in the claimed benefits.

**Additional Comments On Reviewer Discussion:**

At the end of the discussion phas, all reviewers still aligned on a negative stance.

---

### Decision · Program_Chairs · 2025-01-22

Reject